# Development and Characterization of Novel Orthodontic Adhesive Containing PCL–Gelatin–AgNPs Fibers

**DOI:** 10.3390/jfb13040303

**Published:** 2022-12-16

**Authors:** Qihan Yuan, Qianqian Zhang, Xuecheng Xu, Yuqing Du, Jidong Xu, Yu Song, Yuanfei Wang

**Affiliations:** 1School of Stomatology of Qingdao University, Qingdao 266003, China; 2Department of Orthodontics, Qingdao Stomatological Hospital Affiliated to Qingdao University, Qingdao 266001, China; 3Department of Central Laboratory, Qingdao Stomatological Hospital Affiliated to Qingdao University, Qingdao 266001, China

**Keywords:** PCL–gelatin–AgNPs fibers, orthodontic adhesive, antibacterial property, tensile bonding strength, *Streptococcus mutans*

## Abstract

Enamel demineralization around brackets is a relatively common complication of fixed orthodontic treatment, which seriously affects the aesthetics of teeth. In this study, a novel orthodontic adhesive containing polycaprolactone–gelatin–silver nanoparticles (PCL–gelatin–AgNPs) composite fibers was prepared to prevent enamel demineralization of orthodontic treatment. First, PCL–gelatin–AgNPs fibers film prepared by electrospinning was made into short fibers and added to traditional orthodontic adhesives (Transbond XT, 3M Unitek) in three different ratios to design a series of composite adhesives containing antibacterial materials. The antimicrobial performance of the control product and the three samples were then evaluated by bacterial live/dead staining, colony-forming unit (CFU) counts, tensile bond strength (TBS), and adhesive residue index (ARI) scores. The composite adhesives’ antimicrobial properties increased with the increasing content of PCL–gelatin–AgNPs short fibers. The addition of complex antimicrobial fibers to 3M Transbond XT adhesive can significantly reduce the CFU of bacterial biofilms (*p* < 0.05). The bacterial survival rate on the surface of the specimen decreased with the increase of PCL–gelatin–AgNPs short fibers (*p* < 0.05). The TBS and ARI values (*n* = 10) indicated that adding PCL–gelatin–AgNPs short fibers had no significant adverse effect on adhesion. Therefore, adding PCL–gelatin–AgNPs short fibers makes it possible to fabricate orthodontic adhesives with strong antibacterial properties without compromising the bonding ability, which is essential for preventing enamel demineralization around the brackets.

## 1. Introduction

Currently, fixed orthodontic treatment is widely used in clinical practice because of its high efficiency and precision [1,2]. However, due to the wearing of the fixed appliance, the intraoral environment changes, and the difficulty of brushing teeth increases, resulting in the accumulation of plaque around the appliance and the metabolic production of acid [3], which in turn causes demineralization of the enamel, which is clinically referred to as “enamel demineralization” (white spot lesion, WSL) [4,5]. Enamel demineralization has become one of the more common complications in fixed orthodontic treatment, with a high incidence of 15–81% [6,7]. *Streptococcus mutans* is a Gram-positive bacterium present in various biofilms on tooth surfaces, and its increased numbers increase the risk of enamel demineralization [8].

Fluorine-based products have been the gold standard for hard tissue remineralization due to their integration into tooth enamel to convert hydroxyapatite from fluorapatite. However, Lanteri et al. found that the use of preorthodontic treatment of fluoride paint or varnish produces lower bond strength, which may increase the risk of brackets falling off [9]. Orthodontic brackets are fixed onto the tooth surface with adhesive. Studies show that plaque is more likely to accumulate on the adhesive surface than on the bracket surface [10]. Sukonapatipark et al. found that bacteria tend to accumulate in a 10 μm wide gap around the bracket floor and the junction with tooth enamel [11]. Orthodontic adhesives can effectively prevent the occurrence of enamel demineralization if they have long-lasting antimicrobial properties [12,13,14]. Therefore, developing orthodontic adhesives with antibacterial properties is necessary to reduce demineralization in susceptible sites. Previous studies incorporated antibacterial materials such as fluorides [15,16], quaternary ammonium compounds [17], titanium dioxide [18], and silver nanoparticles [19] into orthodontic adhesives to prepare novel adhesives with good antibacterial effects. Therefore, how to give orthodontic adhesive good antibacterial properties without affecting its adhesion has become one of the hot spots in the field of orthodontics in recent years [20,21].

Gelatin is a hydrolyzate of collagen [22] and is highly biocompatible and biodegradable in physiological environments [23,24]. Due to its strong controlled release potential, it has a broad application prospect in drug delivery systems, which can be applied to the in vivo delivery of antibacterial, anti-inflammatory, and antitumor drugs [25,26,27]. In recent years, nanomaterials have been widely used in many disciplines, some of which are pretty amazing in terms of bactericidal range, ability, and persistence [18], such as nanosilver, which can change the permeability of bacterial cell membranes, causing the loss of nutrients inside the bacteria, resulting in bacterial death [28]. Polycaprolactone (PCL) is a synthetic polymer material with good biocompatibility and degradability [29] that was approved by the United States Food and Drug Administration (FDA) as a medical polymer that can be used as a biomaterial [30]. Currently, PCL is widely used in the field of medicine, such as for controlled-release drug carriers, surgical degradation sutures used in surgery, etc.

In this study, we synthesized a material with excellent biocompatibility and long-lasting antimicrobial properties and added it to a resin adhesive to prevent enamel demineralization during orthodontic treatment. In order to improve the performance of the new antibacterial resin adhesive, PCL and gelatin were used as carriers, and the different proportions of antibacterial agent nanosilver were added to the orthodontic adhesive by electrospinning method, thus improving the antibacterial performance. The present study provides a new idea and method for preventing enamel demineralization around the brackets of fixed appliances, which has clinical practical value and application prospects. The null hypothesis of this study is as follows: orthodontic adhesives incorporating PCL–gelatin–AgNPs short fibers (1) do not have acceptable biocompatibility, (2) do not have antimicrobial properties, and (3) do not have good mechanical properties.

## 2. Materials and Methods

### 2.1. Preparation of PCL–Gelatin–AgNPs Composite Fibers

Polycaprolactone (PCL) (Sigma Aldrich, St. Louis, MO, USA; CAS: 24980-41-4) and gelatin (Sigma Aldrich, St. Louis, MO, USA; CAS:9000-70-8) were mixed 1:1 by mass and added to 10 mL of hexafluoroisopropanol (Solarbio, Beijing, China) solvent for a final mass concentration of 12% [31]. AgNPs (Aladdin, Los Angeles, S.C., USA) were added to the above solutions at final concentrations of 0%, 1%, 5%, 10%, and 15%, respectively, and stirred magnetically at 1000 r/min for 8 h to obtain 5 groups of mixed solutions. The above spinning solution was sucked into a disposable syringe, and a blunt needle (type 21G) was attached to the syringe. The high-voltage electrostatic generator was connected to the needle through electrodes, and aluminum foil was fixed to the drum of the homemade receiving device and connected to the ground of the receiving device. Under high voltage, the spinning solution was sprayed in an electric field and deposited on the receiver to form a fiber film. The receiving distance during spinning was set to 15 cm, the spinning voltage and flow rates were 15 kV and 1 mL/h, respectively, the temperature was 23 °C, and the relative humidity was about 30% [31].

### 2.2. Scanning Electron Microscope (SEM) to Observe the Morphology of Composite Fibers

The surface morphology of the fibers was observed with a Gemini SEM 300 instrument (Carl Zeiss Jena, Jena, Germany) with an accelerating voltage of 3 kV; the diameter of the fibers was measured with ImageJ image analysis software, more than 50 fibers were taken from each sample, and the fiber diameters were calculated.

### 2.3. X-ray Diffraction Analysis

The presence of silver nanoparticles (AgNPs) was determined with an X-ray diffraction analyzer (Rigaku Corporation, Tokyo, Japan). The test speed was 10°/min, and the range was 10–90°.

### 2.4. Contact Angle Detection

A drop of water (2 μL) was deposited on the membrane surface, and the contact angle of the fiber membrane surface was measured using a video contact angle system (FTA100, First Ten Angstroms, St. Portsmouth, VA, USA). The hydrophilicity of the composite fiber membrane was evaluated by comparing the size of the contact angle difference.

### 2.5. Agar Diffusion Experiment

The strain of *Streptococcus mutans* ATCC 25,175 (Shanghai Bioresources Collection Center, Shanghai, China) used was cultured overnight in brain heart infusion (BHI) at 37 °C in a 5% CO_2_ atmosphere [32]. A total of 100 μL of *Streptococcus mutans* (ATCC 25175) suspension with a colony count of 106 colony-forming units (CFU)/mL was spread evenly on the BHI nutrient agar plate medium. The composite fiber samples were placed on top and incubated in a constant temperature incubator at 37 °C for 24 h. The diameters of the inhibition zone around the fiber samples were then measured with a vernier caliper three times in different directions, and the average value was taken as the diameter of the inhibition zone of the sample.

### 2.6. Biocompatibility Testing

The nanofiber membranes were cut into circles of 8 mm in diameter with a punch and placed in 24-well plates. The fibrous membranes were sterilized by irradiation under ultraviolet light and rinsed 3 times with phosphate-buffered saline (PBS), after which 400 µL of Dulbecco’s Modified Eagle Medium containing 10% fetal bovine serum (FBS) and 10^4^ L929 cells were added to the wells. The same number of cells and medium were added to the blank well as a control. On days 1, 3, and 5 of cell culture, the old medium was removed, and 400 μL of fresh medium containing 10% cell counting kit-8 was added to each culture well after washing with PBS. After incubation at 37 °C for 2 h, 100 μL of culture solution in each well was transferred to a 96-well plate in 3 parallel for each sample after gentle shaking and mixing well. The absorbance values at 450 nm wavelength were detected by a microplate reader, and the proliferation of cells in each group was compared.

### 2.7. Preparation of Composite Orthodontic Adhesive

The fibrous membranes were removed after vacuum-drying and cut into pieces (1~1.5 cm^2^) with scissors. The cut fiber membrane pieces were dispersed in a specific concentration of polyvinyl alcohol [33], after which the fibrous membranes were homogenized and crushed using a homogenizer. The obtained homogeneous solution was washed and centrifuged many times (not less than 3 times), and the final precipitate was taken, dispersed in water, frozen at −80 °C for 2 h, and vacuum-dried in a freeze dryer for 3 days to obtain composite nanofiber powder [34].

The fibers were mixed with the control group 3M Transbond XT paste (3M Unitek, Monrovia, CA, USA) according to the mass fraction of 1%, 3%, and 5%, placed into the circular mold that was placed in the 96-well plate cover in advance, and applied forcefully. The excess adhesive was extruded and scraped off, and a curing light (Optilux 50; Kerr, Danbury, CT, USA, light intensity 650 mW/cm^2^) was used on the glass side for 60 s to cure fully, obtaining specimens approximately 8 mm in diameter and 1 mm in thickness. The cured flat sample was removed, and the thin surrounding edges were polished off with a slow-speed handpiece. After the specimen was removed from the mold, the disk was magnetically stirred in distilled water at 100 rpm for 1 h to remove uncured monomers [35,36]. The specimen was soaked in 75% alcohol for 3 min and placed on a clean bench; both sides were irradiated by ultraviolet light for 30 min for standby.

### 2.8. CFU Count

After coculturing the samples with 1 mL of *Streptococcus mutans* solution at a concentration of 1 × 10^6^ CFU/mL for 6 h, the amount of bacterial adhesion on different samples was quantitatively analyzed by CFU. The samples were rinsed twice with PBS, and the disc-shaped samples containing biofilms were transported to a test tube containing 1 mL of cysteine peptone water and sonicated for 3 min to fully collect the bacteria on the surface of the samples [35,37]. The bacterial liquid was serially diluted at different times, and 20 μL of the bacterial liquid dilution of each sample was inoculated on an agar plate and incubated aerobically at 37 °C for 24 h. The CFU calculation was based on the colony count and multiplied by the dilution factor, and the formula is shown below.

CFU/mL = the average number of colonies in three replicates of the same dilution × dilution ratio.

### 2.9. Bacterial Live/Dead Staining Test

The composite fibers were placed in 12-well plates, sterilized under UV light, and 1 mL of bacterial solution with a concentration of 1 × 10^7^ CFU/mL was added to the well plate. They were incubated at 37 °C for 48 h, then rinsed 3 times with sterile PBS solution. The composite fibers were stained with live/dead fluorescent staining reagent (Solarbio, Beijing, China) and placed in the dark at room temperature for 20 min. Live/dead fluorescent staining reagents included green fluorescent DNA stain (NucGreen) for various cells and red fluorescent DNA stain (EthD-III) for membrane-damaged cells. Images were observed using a Zeiss laser scanning confocal microscope (LSM 710), and bacterial cells with intact membranes were stained green, while cells with damaged membranes were stained red.

### 2.10. Adhesive Performance Experiment of Composite Antibacterial Resin Adhesive

To evaluate the adhesion properties of the antimicrobial adhesive, human premolars were used for the study. This study was conducted according to the Qingdao Stomatological Hospital Ethics Committee protocol (2020KQYX031), and consent was obtained from all participants in the study. From October 2020 to February 2021, 40 premolars with intact enamel and standard color were selected, and 10 teeth were tested for each group. Stainless steel premolar double brackets (Xinya dental material, Hangzhou, China) with a base plate area of 12.97 mm^2^ were used in this experiment.

Before bracket bonding, the tooth was brushed thoroughly with fluoride-free toothpaste, rinsed under running water, and polished with a silicone polisher and low-speed handpiece for 10 s. Then, it was rinsed with distilled water, blow-dried, and etched with 37% phosphoric acid (TransbondTM XT etching gel; 3M Unitek, Monrovia, CA, USA) at the center of the clinical crown of the isolated tooth. The teeth were acidly etched for 30 s, pressure rinsed for 30 s, and then surface dried, with a chalky appearance of the etched portion. A thin layer of primer (TransbondTM XT Light Cure Orthodontic Primer; 3M Unitek, Monrovia, CA, USA) was applied with a small cotton swab, and an appropriate amount of the new adhesive was placed on the base of the bracket and placed in the center of the buccal clinical crown of the isolated teeth. After a slight pressure to remove excess adhesive around the bracket, a light curing lamp (Optilux 50; Kerr, Danbury, CT, USA, light intensity 650 mW/cm^2^) was placed on the gingival, occlusal, and near brackets, for intermediate and distal curing for 10 s [17].

To simulate oral aging conditions, the sample was soaked in artificial saliva and replaced every three days. After one month of soaking, the teeth were embedded in a superhard plaster of 20 mm× 7 mm × 7 mm, exposing the buccal tooth surface. The angle of the isolated tooth was adjusted before the plaster was hardened so that when the bracket was stressed, the direction of the shear force was perpendicular to the horizontal groove of the bracket. Each group produced 10 specimens for subsequent bond strength and ARI index evaluation.

A universal tensile machine (WDW-5T) was used to test the bond strength. The base of the sample was firmly fixed with the clamp on the tensile machine, and 0.019 × 0.025 inch stainless steel wire was placed into the groove of the bracket and fixed to the movement arm of the tensile machine with ligature wire [38]. The moving component of the tensile machine was pulled upward at a constant speed of 0.5 mm/s [39], and the tensile machine recorded the maximum tensile force value received by the specimen when it was damaged.
(1)$$\mathrm{Bonding}\;\mathrm{strength}\;(\mathrm{Mpa})=\frac{\mathrm{Maximum}\;\mathrm{load}\;\left(N\right)}{\mathrm{Bottom}\;\mathrm{area}\;\mathrm{of}\;\mathrm{bracket}\;\mathrm{mesh}\;\left({\mathrm{mm}}^{2}\right)}$$

The adhesive residue index (ARI) developed by Artun and Bergland was assessed to quantify the adhesive residue on the enamel surface after the brackets fell off [16,40]. ARI scores were recorded under a 10× optical microscope. The ARI (matrix ARI score or ARI) score is 0–3 as follows:

0: No adhesive residue on the teeth;

1: Less than 50% adhesive remains on the teeth;

2: More than 50% of the adhesive remains on the teeth;

3: All the adhesive remains on the teeth.

### 2.11. Statistical Analysis

SPSS version 21 (SPSS, New York, NY, USA) was used for statistical analysis of the results. Single-factor analysis of variance (ANOVA) and least significant difference *t*-test (LSD-t) were used to compare the differences among groups, and the chi-square test was used to evaluate ARI scores. The statistical significance was set as *p* < 0.05.

## 3. Results

### 3.1. SEM Image of PCL–Gelatin–AgNPs Fibers

Figure 1a–d show the SEM photos of electrospun PCL–gelatin–AgNPs fibers, which showed a decreasing trend when 1%, 5%, 10%, and 15% of nanosilver was added, with the average diameter of 0.95 μm, 0.94 μm, 0.84 μm, and 0.83 μm, respectively, in a normal distribution (Figure 1e–h). The surface of each group of fibers was smooth, without beading or adhesion.

### 3.2. X-ray Diffraction Analysis

Figure 2 shows the XRD pattern of the PCL–gelatin–AgNPs fibers. The nanosilver particles had prominent absorption peaks, which were consistent with the data on JCPDS card 04-0783 at 38.096°, 44.257°, 66.406°, and 77.452°, respectively [41,42]. Corresponding to the (111), (200), (220), and (311) planes of cubic silver, respectively, the existence of AgNPs in the fibers was confirmed.

### 3.3. Contact Angle Test

The fiber membranes with different AgNPs contents had certain hydrophilicity. Figure 3a–d reflect the contact angle images of the composite nanofiber membranes when the AgNPs content was 1%, 5%, 10%, and 15%, whose contact angles were 55.5°, 58.2°, 59.1°, and 63.3°, respectively. When the water contact angle of the material was between 0° and 90°, it can be called wettability. It can be seen that the fiber membranes with different AgNPs additions exhibited hydrophilic properties, which facilitate further bonding with the adhesive at a later stage.

### 3.4. Agar Diffusion Experiment

As shown in Figure 4, the agar diffusion method compared the in vitro antibacterial effects of electrospun fibers with different AgNPs contents against *Streptococcus mutans*. It was found that the gelatin nanofiber membrane without AgNPs had no antibacterial properties, and bacteria grew on the samples without an inhibition zone. The diameter of the antibacterial ring gradually increased with the addition of AgNPs.

### 3.5. Biocompatibility Testing of Composite Fibers

Figure 5 shows the proliferation of L929 cells on each composite fiber. It can be observed from the figure that the cells were well propagated and differentiated on each fibrous membrane. The OD values of electrospun fiber groups with different AgNPs contents had no statistical difference (*p* > 0.05). The results indicated that although the composite fibers had specific cytotoxicity, they still ensured cell proliferation.

### 3.6. CFU Counts of Antibacterial Adhesives

The results of CFU counts of *Streptococcus mutans* for the 8 h biofilms on discs are shown in Figure 6a. The addition of composite antibacterial fibers in 3M Transbond XT adhesive effectively reduced the total number of bacteria. The CFU counts on composite discs containing 5% PCL–gelatin–AgNPs fibers were two orders of magnitude lower than those in the Transbond XT control group (*p* < 0.05). Figure 6b(A–D) show the CFU counts of biofilms from 3M Transbond XT binder, 1% AgNPs fiber, 3% AgNPs fiber, and 5% AgNPs fiber discs after 10^7^ dilutions over 8 h. It can be seen that the number of colonies decreased to different degrees with the increase of AgNPs content in the fibers.

### 3.7. Bacterial Live/Dead Staining Test

Figure 7a(A–D) show the staining plot of live dead bacteria growing for 2 days on the surface of PCL–gelatin–AgNPs staple fibers with 0%, 1%, 3%, and 5% additions. The Transbond XT control group was covered by a biofilm consisting mainly of live bacteria, and the total number of live bacteria gradually decreased with increasing content of the composite antimicrobial fiber. As shown in the figure, the surface of the adhesive containing 5% AgNPs had a large number of red-stained dead bacteria. Figure 7b shows that the bacterial survival rate of these four adhesives decreased with the crease of the antimicrobial material, and the differences in the rate of live bacteria in the biofilm of each group were statistically significant (*p* < 0.05).

### 3.8. Adhesive Properties of Antibacterial Adhesives

The TBS values of orthodontic adhesives are shown in Table 1. The TBS value of the adhesive gradually decreased with the increase of the antibacterial material. However, the TBS values of the three experimental groups were around 9.5 Mpa, and there was no significant difference between them and the control group (*p* > 0.05). The new antibacterial adhesive materials prepared in this experiment all meet the bond strength requirements, indicating that the addition of the appropriate amount of antimicrobial substances did not affect the adhesive performance or change the physical properties of the adhesive. Table 2 shows the ARI scores of the orthodontic adhesive, indicating the amount of adhesive remaining after the TBS test. The ARI scores of the four groups were mainly distributed between 1 and 2, indicating that the bonding failure mainly occurred between the adhesives. There was no significant difference between the groups (*p* > 0.05).

## 4. Discussion

Fixed orthodontics leads to more difficult oral hygiene maintenance for patients, which causes plaque accumulation around brackets and significantly increases the risk of enamel demineralization [43]. To reduce the incidence of enamel demineralization, this study was the first to incorporate PCL–gelatin–AgNPs fibers as an antibacterial material into a conventional orthodontic bonding adhesive, giving it antibacterial properties. Based on these results, it was confirmed that the composite fibers had acceptable biocompatibility on day 5. It was further deduced that the new bonding adhesive with a small number of short fibers had acceptable biocompatibility, and therefore the first null hypothesis was rejected. In addition, the new adhesive had good antibacterial and mechanical properties, and the second and third null hypotheses were rejected.

Orthodontic adhesive fixes brackets or bands to the teeth without patient cooperation and is particularly suitable for adolescents with poor compliance. Therefore, developing an orthodontic adhesive with antibacterial properties is necessary to reduce demineralization in susceptible areas [12]. Nanosilver has an excellent performance in terms of bactericidal ability and durability [18], and has a broad application prospect. The specific antibacterial mechanism of AgNPs is not conclusive, but the leading theories are (a) destruction of the bacterial wall; (b) generation of reactive oxygen species, which react with bacteria in an oxidative manner; (c) destruction of the active structure on the surface of the bacterium; and (d) entry into the cell to inactivate enzymes and denature proteins [44]. Due to the good antimicrobial properties of nanosilver, we believe that adding composite nanosilver materials can enhance the antimicrobial properties of the orthodontic adhesive.

Agar diffusion tests confirmed the antimicrobial properties of the PCL–gelatin–AgNPs fibers. It was found that there was no significant difference in the size of the inhibition rings presented by fibers with 10% and 15% of AgNPs added. In order to avoid cost increases and waste of resources due to the excessive addition of AgNPs, the 10% AgNPs-PCL composite fiber will be applied for subsequent experiments. In addition, the effect of the novel adhesive on the growth and metabolic activity of *Streptococcus mutans* was assessed by CFU counts and bacterial live/dead staining experiments. The results consistently showed that bacterial growth was significantly inhibited by adding AgNPs-loaded fibers to the adhesive. Therefore, the second null hypothesis of this study was rejected.

There are studies where Ag nanoparticles were recently incorporated into resins [19,45,46]. The small particle size and large surface area of AgNPs allow them to release more Ag^+^ at low filler levels, thereby reducing the concentration of Ag^+^ required for efficacy [47,48]. Li et al. found that adhesives containing AgNPs inhibited *Streptococcus mutans* on their surface and *Streptococcus mutans* suspended in the medium away from their surface [49]. This suggests that resins containing AgNPs are long-range lethal, possibly due to the release of Ag^+^ [50]. Previous studies show that Ag-containing resin composites have long-lasting antimicrobial activity due to the continuous release of Ag^+^ [51]. Yoshida et al. found that Ag-containing dental composites were shown to inhibit the growth of *Streptococcus mutans* when tested for 6 months [52].

The ideal orthodontic adhesive should have the mechanical strength to prevent brackets dislodgement during orthodontic procedures. Orthodontic brackets are affected by a variety of forces in complex oral environments, and studies show that tensile and shear strength can better reflect bond strength to a certain extent [53,54]. Studies found that with the increase of antimicrobial materials, resin adhesives may experience a decrease in mechanical properties [55]. Reynolds recommends a minimum bond strength of 5.9–7.8 Mpa to meet bond strength requirements [56]. However, bond strength is not as strong as possible, and it was reported that bond strength above 13.5 Mpa can cause damage to enamel during debonding [57]. Although the orthodontic adhesive TBS decreased slightly with the increase of composite fibers in this study, no significant difference was found when the experimental group was compared with the control group. The TBS values in each group were higher than 5.9 MPa and lower than 13.5 Mpa, all of which met the clinical adhesion criteria. Therefore, the third null hypothesis was not rejected.

There are three kinds of bracket shedding interfaces: one is the interface fracture, the section is located between the enamel and the adhesive, which easily causes damage to the enamel surface. The second is cohesion failure, and the fracture surface is located between the adhesive. The third is the destruction between the bracket and the adhesive; this fractured form will not harm the enamel, but the adhesive material has more residue on the tooth surface, which increases the chairside operation time [58]. The second and third fracture forms are ideal. The lower the ARI score, the simpler the cleaning of the adhesive after the bracket is detached, and the destruction interface tends to occur between the enamel and the adhesive. The higher the ARI score, the more adhesive material remains on the tooth surface, and the fracture tends to occur between the adhesive and the bracket [59]. In this study, the ARI scores between the adhesives in the experimental and control groups were mostly concentrated in one and two points, indicating that the bond failure mostly occurred between the adhesives. However, the ARI index is mainly subjective, and it is not easy to distinguish the area of debonded material on the tooth and the bracket floor [60]. At the same time, considering the potential oral microenvironment and occlusal stress, the current experiment is inaccurate in simulating the actual situation in the oral cavity. Therefore, further clinical studies are essential.

If antibacterial materials can be combined with materials that promote enamel remineralization, the incidence of enamel demineralization is significantly reduced. In recent years, a bioactive glass for dentistry was developed and applied to numerous studies to remineralize WSL [61,62,63]. Bioactive glass shows more effective and rapid enhancement of tooth enamel remineralization than milk-protein-derived casein phosphopeptide-amorphous calcium phosphate (CPP-ACP) [64]. Scribante et al. used biomimetic nanohydroxyapatite to induce visible tooth enamel remineralization after two application cycles, and the bond strength can be considered clinically acceptable [65]. In addition, some scholars added zinc–hydroxyapatite with remineralization to toothpaste and resin composite, both of which show good enamel protection [66,67,68]. Therefore, combining antimicrobial substances and remineralized materials is a strategy to prevent WSL.

## 5. Conclusions

In this study, composite PCL–gelatin–AgNPs fibers were prepared. Biocompatibility results showed that the PCL–gelatin–AgNPs fibers had no cytotoxicity, and the agar diffusion test showed good antibacterial activity. The composite PCL–gelatin–AgNPs short fibers combined with clinical orthodontic adhesions showed good antibacterial properties without affecting the bonding properties, which provides a new idea for clinical orthodontic prevention of demineralization.

## Figures and Tables

**Figure 1 jfb-13-00303-f001:**
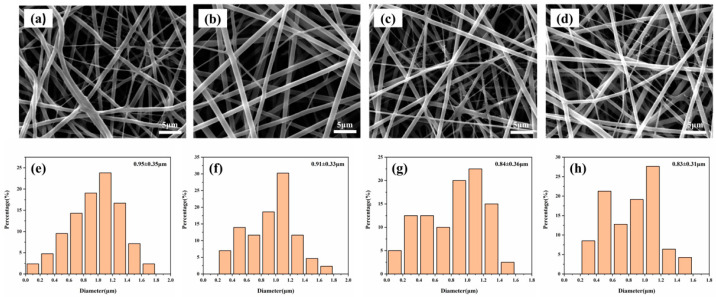
(**a**–**d**) The SEM photographs of PCL–gelatin–AgNPs fibers at 1%, 5%, 10%, and 15% nanosilver content, and (**e**–**h**) the corresponding fiber diameter distribution and frequency distribution.

**Figure 2 jfb-13-00303-f002:**
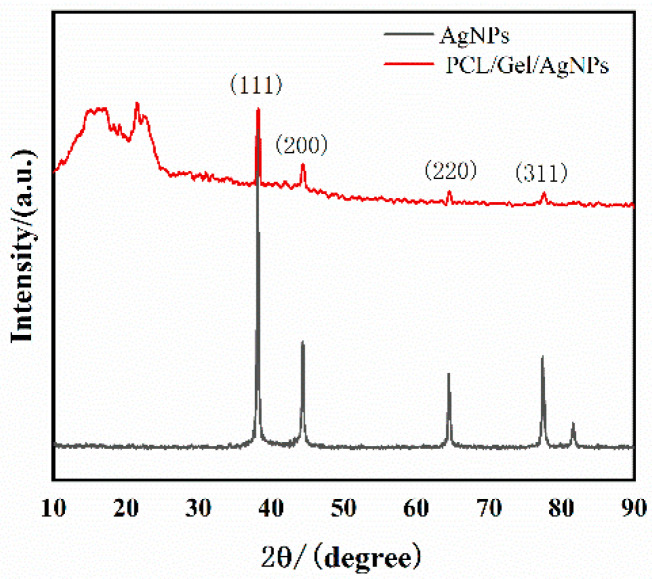
The XRD spectrum of the PCL–gelatin–AgNPs fibers showed that the absorption peak of the nanosilver particles confirmed the existence of AgNPs in the fiber.

**Figure 3 jfb-13-00303-f003:**
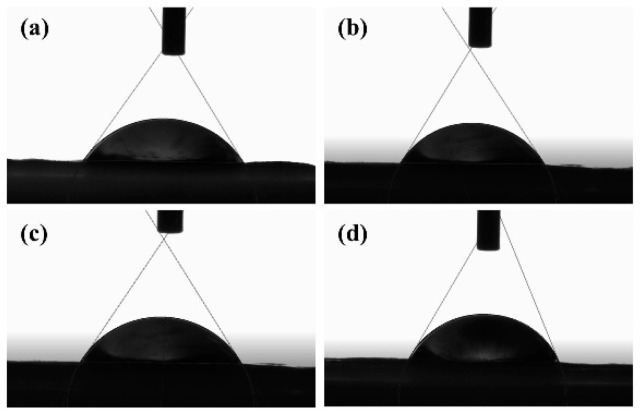
(**a**–**d**) The contact angles of the composite nanofiber for AgNPs additions were 1%, 5%, 10%, and 15%, respectively.

**Figure 4 jfb-13-00303-f004:**
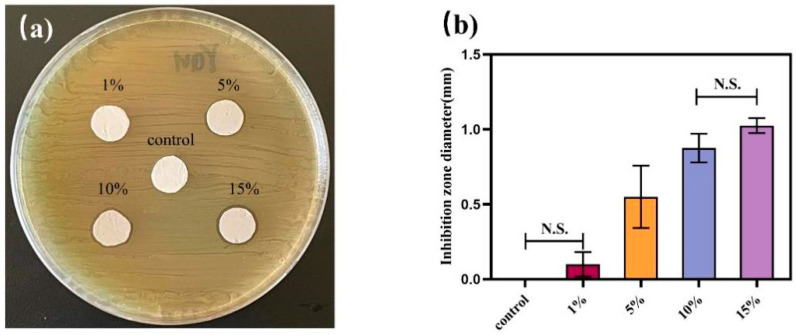
The antibacterial effects of composite fibers with different AgNPs contents against *Streptococcus mutans* were determined by the agar plate diffusion method. (**a**) Photographs of antibacterial rings formed after cocultivation of fiber membranes with *Streptococcus mutans* at different nanosilver contents for 24 h. There was no obvious antibacterial ring formation for composite fibers with AgNPs content of 1%, and the rings of fibers inhibiting bacteria gradually increased with the increase of AgNPs content. (**b**) The difference in the size of the inhibition ring: PCL/gel < PCL/gel/1%AgNPs < PCL/gel/5%AgNPs < PCL/gel/10%AgNPs < PCL/gel/15%AgNPs, compared with the blank and PCL/gel/1%AgNP groups, the PCL/gel/5%AgNPs, PCL/gel/10%AgNPs, and PCL/gel/15%AgNPs groups had statistically significant inhibition effects (*p* < 0.05).

**Figure 5 jfb-13-00303-f005:**
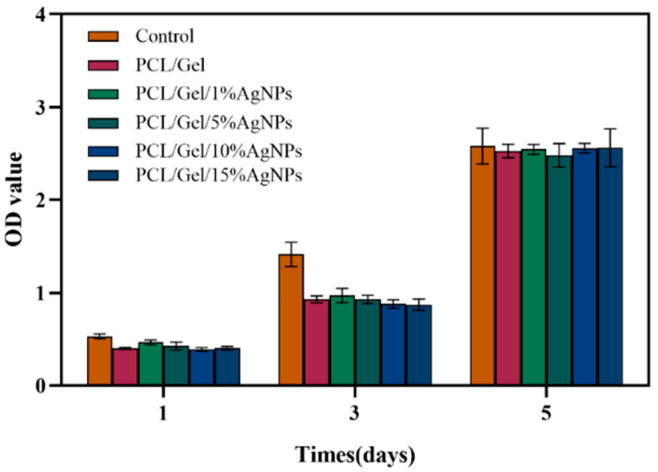
The proliferation of L929 cells on different fibrous membranes was detected by the CCK-8 method. From the 1st to the 5th day, the number of cells in each group increased rapidly, and on day 5, there was no significant difference between the OD values of the control group and each experimental group (*p* > 0.05).

**Figure 6 jfb-13-00303-f006:**
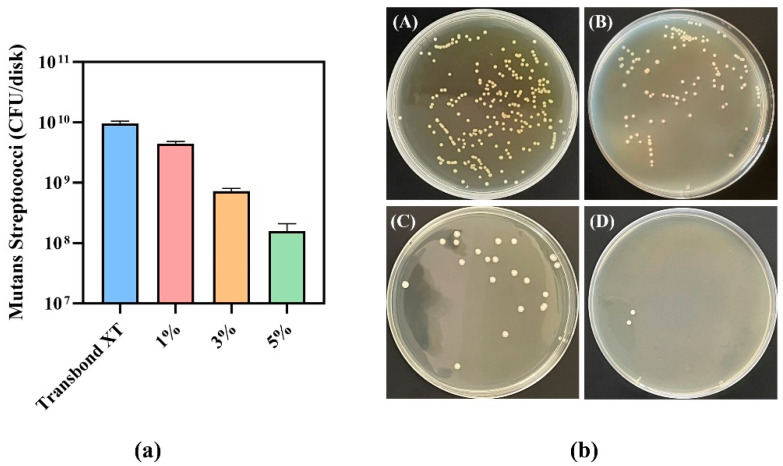
(**a**) CFU counts of biofilms cultured on disc for 8 h. The addition of composite antibacterial fibers in 3M Transbond XT adhesive can reduce the CFU of biofilms. There were significant differences (*p* < 0.05) in CFU between groups with different antibacterial material addition amounts. (**b**) A–D are the CFU counts of biofilms cultured on 3M Transbond XT adhesive, 1% AgNPs fiber, 3% AgNPs fiber, and 5% AgNPs fiber discs after 10^7^ dilutions for 8 h. Under the same dilution factor, the number of colonies decreases to varying degrees with the increased AgNPs content in fibers.

**Figure 7 jfb-13-00303-f007:**
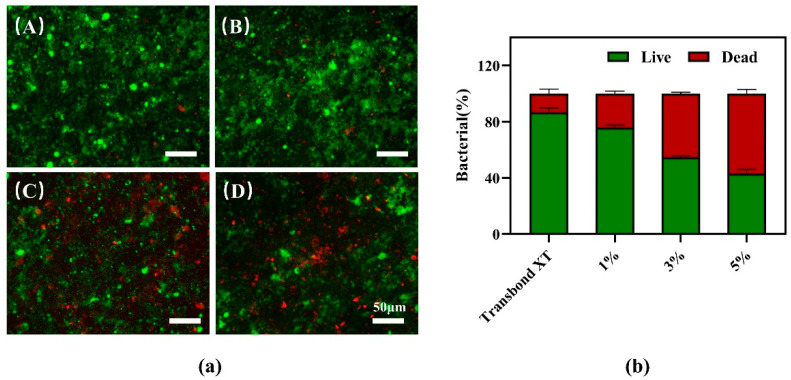
(**a**) Staining plot of live dead bacteria growing on the surface of PCL–gelatin–AgNPs in amounts of 0% (A), 1% (B), 3% (C), and 5% (D) for 2 days, with the live bacteria in green and the dead bacteria in red. (**b**) The proportion of live and dead bacteria on the resin discs. The control disc was covered with live bacteria, while most bacteria on the 5% PCL–gelatin–AgNPs resin disc died.

**Table 1 jfb-13-00303-t001:** The TBS value of orthodontic adhesives.

Groups	Number	TBS (Mpa)				*p* Value
		Mean	Standard Deviation	Min	Max	
Transbond XT	10	10.53	1.47	8.31	12.34	-
Transbond XT + 1% PCL–gel–AgNPs	10	9.95	1.09	7.95	11.50	0.674
Transbond XT + 3% PCL–gel–AgNPs	10	9.57	1.15	7.88	11.29	0.256
Transbond XT + 5% PCL–gel–AgNPs	10	9.36	0.73	7.54	10.25	0.118

**Table 2 jfb-13-00303-t002:** The orthodontic ARI scores of orthodontic adhesives. Groups with the same letter (a) are not statistically different (*p* > 0.05).

Groups	ARI Scores				Statistical Difference
	0	1	2	3	
Transbond XT	2	3	3	2	a
Transbond XT + 1% PCL–gel–AgNPs	0	5	4	1	a
Transbond XT + 3% PCL–gel–AgNPs	1	3	5	1	a
Transbond XT + 5% PCL–gel–AgNPs	1	3	4	2	a

## Data Availability

Not applicable.

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
