# Peer review of "Development and Characterization of Novel Orthodontic Adhesive Containing PCL–Gelatin–AgNPs Fibers"

_jfb, 2022, doi:10.3390/jfb13040303_

Round 1
Reviewer 1 Report
This article is interesting because of developing the new anti-bacterial adhesive cement for orthodontic bracket setting. The scope of this Journal will take in the contents of this manuscript.
However, this reviewer possesses some questions and suggests some corrections.
Questions:
In the introduction section, the authors emphasized lacking sustainable anti-bacterial effect of previously developed anti-bacterial orthodontic adhesive. Whereas the protocol of this study did not show the long-term anti-bacterial effect of the new resin adhesives containing Geltin-AgNPs composite fibers. Will the authors present the long-term anti-bacterial effect in the next step experiment?
In the methodology, the authors examined the anti-bacterial effect and biocompatibility of Geltin-AgNPs composite fibers; however, this examination is unnecessary because the composite fibers exist inside of the polymerized resin adhesive. Why did the authors examine the anti-bacterial effect and biocompatibility of Geltin-AgNPs composite fibers. Moreover, the authors did not examine the biocompatibility of the polymerized the resin adhesives containing Geltin-AgNPs composite fibers. This reviewer considers examination of the resin adhesive biocompatibility is more important.
The authors prepared the composite fibers film using electrospinning method and crushed to make the composite nanofiber powder. Did the authors confirm the length of the composite nanofiber particle? The authors had better show the SEM image of the particle.
Suggestions:
In the Title, “AgNPs Fibers” had better change to “PCL-Gelatin-AgNPs fibers”.
Line 15-16
“Polycaprolactone-Gelatin-Silver nanoparticles (PCL-Gelatin-AgNPs) composite fibers” could be better.
Line 16
“PCL-Gelatin-AgNPs fibers film” may be better.
Line 76
“Polycaprolactone (PCL)” is better.
Line 77
“Hexaflouroisopropanol” is better.
Line 80
The authors described “5 groups of mixed solutions named A-E.”; however, no named groups appeared.
Line 97
"silver nanoparticles (AgNPs)" is better.
Line 108
“colony-forming unit (CFU)/mL” is better.
“brain heart infusion (BHI)” is better.
Line 117
UV had better change to ultra violet.
“phosphate-buffered saline (PBS)” is better.
“Dulbecco's Modified Eagle Medium” is better.
Line 118
What is “FBS”? This may be “PBS”.
Line 129
“cell counting kit-8” is better.
Line 137
The authors used 96-well plate to make the adhesive disks. There was no description about the disk size. The authors had better describe the diameter and height of the disk.
Line 196-197
“with the ligation wings up,” should be eliminated.
Line 344
The reference of #15 should be added such as [15, 45, 46]
Line 362
The reference of #45 should be added such as [15,45, 46]
Line 365-367
The reference number should be added at the end of the sentence.
Line 367
What is “the OCT experiments?
Line 371-373
The reference number should be added at the end of the sentence.
Line 381
“studies” should be changed to “the study” because of single reference.
Line 382-384
The two sentences need addition of reference number(s) at the end of each sentence.
In the Discussion section, the authors had better discuss concerning the mechanism how Ag ions (anti-bacterial agent) is released from the polymerized adhesive containing PCL-Gelatin-AgNPs fibers. Furthermore, the authors had better measure the amount of Ag ions released from the polymerized adhesive at each time during long term storage to prove the long-term Ag ion release ability of this adhesive in future.
Author Response
Dear reviewer,
Thanks a lot for your kindly comments on our manuscript. These comments are valuable and very helpful for revising and improving our manuscript. In what follows, we would like to answer the questions you mentioned and give detailed account of the changes made to the original manuscript.
Q1. In the introduction section, the authors emphasized lacking sustainable anti-bacterial effect of previously developed anti-bacterial orthodontic adhesive. Whereas the protocol of this study did not show the long-term anti-bacterial effect of the new resin adhesives containing Geltin-AgNPs composite fibers. Will the authors present the long-term anti-bacterial effect in the next step experiment?
Response: Thanks a lot for your question, Due to time, we only conduct short-term antibacterial experiments as follows, and long-term antibacterial effect will be verified in the future.
This experiment carried out the MTT method to detect bacterial biofilm activity pre-experiment, the specific method is as follows: Make 9 adhensive disks with AgNPs-PCL addition of 0%, 1%, 3% and 5%, divided into 3 groups, each group includes 3 adhensive disks mentioned above, soak the adhensive disks in distilled water, replace them every 3 days, soak for 1 day, 14 days, 28 days, respectively, disinfect and reserve. The aged resin pieces were transferred to a sterile 24-well plate and 10 μl of S. mutans bacterial solution at a concentration of 108 cfu/ml and 1 ml of BHI liquid medium were added to each well and incubated at 37°C for 24 h. The 24-well plate was removed and gently rinsed with PBS buffer to remove the loose Streptococcus mutans from the surface of the specimen. Transfer the specimens to a new 24-well plate, add 1 mL of 0.5 mg/mL MTT solution to each well, incubate at 37℃ for 1 hour, then aspirate the MTT solution and add 1 mL of dimethyl sulfoxide (DMSO) to each well. After shaking for 20 minutes, mix well and aspirate 200 μl of bacterial solution in each well into a 96-well plate. 3 parallel controls were set up for each test t piece and he OD540 of each well was measured by enzyme marker.
RBV =×100%
The experimental results are as follows:
After 1 day and 2 weeks of ageing treatment, each antimicrobial fiber incorporation group exhibited relatively low bacterial viability with statistically significant differences (P<0.05), indicating that different concentrations of the new bonding agent exhibited antimicrobial properties. After 4 w of immersion, there was no statistically significant difference in relative biofilm viability between the Transbond XT group and the 1% AgNPs-PCL group (P>0.05), while there was a statistically significant difference in relative biofilm viability between the 3% AgNPs-PCL and 5% AgNPs-PCL groups (P<0.05). This indicates that the antibacterial activity of the new adhesive gradually decreased in all groups as the ageing time of the specimens increased. However, this pre-experiment was performed on only one specimen per concentration and the short ageing time was controversial in terms of experimental rigour and therefore not mentioned in the article.
Q2. In the methodology, the authors examined the anti-bacterial effect and biocompatibility of Geltin-AgNPs composite fibers; however, this examination is unnecessary because the composite fibers exist inside of the polymerized resin adhesive. Why did the authors examine the anti-bacterial effect and biocompatibility of Geltin-AgNPs composite fibers. Moreover, the authors did not examine the biocompatibility of the polymerized the resin adhesives containing Geltin-AgNPs composite fibers. This reviewer considers examination of the resin adhesive biocompatibility is more important.
Response: We agree with the reviewer that further elaborating on this point using new data would be helpful.However, we believe that the PCL-Gelatin-AgNPs staple fibers prepared in this experiment exist not only inside the adhesive, but also on the surface of the adhesive and the contact surface with the teeth, so we performed a biocompatibility experiment on the fiber membrane. As a biocompatible material, orthodontic adhesive has been widely used in clinical practice, so we can consider that the biocompatibility of the synthesized new adhesive is better.
Q3. The authors prepared the composite fibers film using electrospinning method and crushed to make the composite nanofiber powder. Did the authors confirm the length of the composite nanofiber particle? The authors had better show the SEM image of the particle.
Response: Thank you very much for the constructive suggestions. We have recently supplemented our SEM experiments with electrospun PCL-Geltin-AgNPs short fibers. the SEM images are shown in Figure A. Figure B shows the length distribution of the short fibers, the average length of which was measured to be 68.13 µm.
- SEM photographs of the SEM photographs of PCL-Geltin-AgNPs fibers
- Diameter distribution of PCL-Geltin-AgNPs fibres
Suggestions:
- In the Title, “AgNPs Fibers” had better change to “PCL-Gelatin-AgNPs fibers”.
Response: Thanks a lot, we have changed “AgNPs Fibers” to “PCL-Gelatin-AgNPs fibers”.
- “Polycaprolactone-Gelatin-Silver nanoparticles (PCL-Gelatin-AgNPs) composite fibers” could be better.
Response: Thanks a lot, we have changed “PCL-Geltin-AgNPs composite fibers” to “Polycaprolactone-Gelatin-Silver nanoparticles (PCL-Gelatin-AgNPs)”.
- “PCL-Gelatin-AgNPs fibers film” may be better.
Response: Thanks a lot, we have changed “PCL-Geltin-AgNPs fibers” to “PCL-Gelatin-AgNPs fibers film”.
- “Polycaprolactone (PCL)” is better.
Response: Thanks a lot, we have changed “PCL” to “Polycaprolactone (PCL)”.
- “Hexaflouroisopropanol” is better.
Response: Thanks a lot, we have changed “HFIP” to “Hexaflouroisopropanol”.
- 6. The authors described “5 groups of mixed solutions named A-E.”; however, no named groups appeared.
Response: Thanks a lot, we have removed the "named A-E" from the original text.
- "silver nanoparticles (AgNPs)" is better.
Response: Thanks a lot, the statement of “AgNPs” was corrected as "silver nanoparticles (AgNPs)" .
- “colony-forming unit (CFU)/mL” is better.
Response: Thanks a lot, the statement of “CFU/mL” was corrected as “colony-forming unit (CFU)/mL”.
- “brain heart infusion (BHI)” is better.
Response: Thanks a lot, the statement of “BHI” was corrected as “brain heart infusion (BHI)”.
- UV had better change to ultra violet.
Response: Thanks a lot, we have changed “UV” to “ultra violet”.
- “phosphate-buffered saline (PBS)” is better.
Response: Thanks a lot, we have changed “PBS” to “phosphate-buffered saline (PBS)”.
- “Dulbecco's Modified Eagle Medium” is better.
Response: Thanks a lot, we have changed “DMEM” to “Dulbecco's Modified Eagle Medium”.
- What is “FBS”? This may be “PBS”.
Response: FBS is an abbreviation for "Fetal Bovine Serum" and we have changed “FBS” to “Fetal Bovine Serum (FBS)”.
- “cell counting kit-8” is better.
Response: Thanks a lot, we have changed “CCK-8 reagent” to “cell counting kit-8”.
- The authors used 96-well plate to make the adhesive disks. There was no description about the disk size. The authors had better describe the diameter and height of the disk.
Response: Thanks a lot, in this experiment, the adhesive disk is about 8 mm in diameter and about 1 mm thick, and the relevant description has been added in the article.
- “with the ligation wings up,” should be eliminated.
Response: Thanks a lot, we have removed “with the ligation wings up” and change the sentence to “Fix the base of the sample firmly with the clamp on the tensile machine, place 0.019 × 0.025-inch stainless steel wire into the groove of the bracket and fix it to the movement arm of the tensile machine with ligature wire”.
- The reference of #15 should be added such as [15, 45, 46]
Response: Thanks a lot, we have added reference #15.
- The reference of #45 should be added such as [15,45, 46]
Response: Thanks a lot, we have added reference #45.
- The reference number should be added at the end of the sentence.
Response: Thanks a lot,
- What is “the OCT experiments?
Response: We gratefully appreciate for your valuable suggestion. We have adjusted the content of the cited documents in the article and added appropriate references.
- “studies” should be changed to “the study” because of single reference.
Response: Thanks a lot, we have changed “studies” to “the study”.
- The two sentences need addition of reference number(s) at the end of each sentence.
Response: Thanks a lot, we have added references #52 and #38 at the end of both sentences.
- In the Discussion section, the authors had better discuss concerning the mechanism how Ag ions (anti-bacterial agent) is released from the polymerized adhesive containing PCL-Gelatin-AgNPs fibers. Furthermore, the authors had better measure the amount of Ag ions released from the polymerized adhesive at each time during long term storage to prove the long-term Ag ion release ability of this adhesive in future.
Response: We are so grateful for your kind question. With the degradation of PCL, the nano-silver is released from it, thus exerting an antibacterial effect. Testing the release of silver ions is important to demonstrate the long-term antimicrobial effect of the orthodontic adhesive produced in this experiment. We plan to illustrate the performance of the new adhesive in subsequent experiments with ion release tests and longer-term antimicrobial performance tests.
In addition, we add the following to the discussion section on the antimicrobial principle of the new adhesive. “The small particle size and large surface area of AgNPs allows them to release more Ag+ at low filler levels, thereby reducing the concentration of Ag+ required for efficacy[1,2]. Li et al. found that adhesives containing AgNPs inhibited not only Streptococcus mutans on their surface, but also Streptococcus mutans suspended in the medium away from their surface[3]. This suggests that resins containing AgNPs are long-range lethal, possibly due to the release of Ag+[4]. Previous studies have shown that Ag-containing resin composites have long-lasting antimicrobial activity due to the continuous release of Ag+[5]. Yoshida et al. found that Ag-containing dental composites were shown to inhibit the growth of Streptococcus mutans when tested for 6 months[6].”
Morones et al. showed that the bactericidal effect of nanosilver is related to the release of Ag+. The data suggest that rapid Ag+ release occurs when the nanoparticles are first dissolved, but only at level of < 5 µm. No further dissolution occurs and the free Ag+ concentration decreases, possibly due to reduction processes to form Ag+ containing clusters or re-association with the original nanoparticles. This analysis corroborated the presence of micro-molar concentrations of silver ions, which will have a contribution to the biocidal action of the silver nanoparticles[4].
Reference:
- Cheng, L.; Weir, M.D.; Xu, H.H.; Antonucci, J.M.; Kraigsley, A.M.; Lin, N.J.; Lin-Gibson, S.; Zhou, X. Antibacterial amorphous calcium phosphate nanocomposites with a quaternary ammonium dimethacrylate and silver nanoparticles. Dental materials : official publication of the Academy of Dental Materials 2012, 28, 561-572, doi:10.1016/j.dental.2012.01.005.
- Cheng, Y.J.; Zeiger, D.N.; Howarter, J.A.; Zhang, X.; Lin, N.J.; Antonucci, J.M.; Lin-Gibson, S. In situ formation of silver nanoparticles in photocrosslinking polymers. Journal of biomedical materials research. Part B, Applied biomaterials 2011, 97, 124-131, doi:10.1002/jbm.b.31793.
- Li, F.; Weir, M.D.; Chen, J.; Xu, H.H. Comparison of quaternary ammonium-containing with nano-silver-containing adhesive in antibacterial properties and cytotoxicity. Dental materials : official publication of the Academy of Dental Materials 2013, 29, 450-461, doi:10.1016/j.dental.2013.01.012.
- Morones, J.R.; Elechiguerra, J.L.; Camacho, A.; Holt, K.; Kouri, J.B.; Ramírez, J.T.; Yacaman, M.J. The bactericidal effect of silver nanoparticles. Nanotechnology 2005, 16, 2346-2353, doi:10.1088/0957-4484/16/10/059.

Reviewer 2 Report
"Development and Characterization of Novel Orthodontic Adhe- 2 sive Containing AgNPs Fibers".
General comments
· The manuscript is well written and well conducted.
· Experimental design is straightforward, and results are mostly clear. However, some critical information is lacking as described below. There are also several other issues that should be addressed.
Specific comments
Abstract
- The sample size is not clear
Introduction:
Material and Methods:
Please add illustrated diagram for the method used to prepare PCL-Geltin-AgNPs composite fibers and provide the CAS number for the utilized materials.
Add suitable references at the end of line 88
Results:
· In Fig. 6B please add a labelling for the 4 plates and in the figure legend add the description of the four plates
· In Fig. 7A also please add a proper labelling for the 4 Confocal microscopy images and in the figure legend to clarify the difference between them.
Discussion:
Please expand the discussion section to include comparing the obtained results to up to date research work of using the bioactive glasses in orthodontics for protecting and remineralizing enamel.

Author Response
Dear reviewer,
We would like to thank you for your careful reading, helpful comments, and constructive suggestions, which has significantly improved the presentation of our manuscript.
Q1. The sample size is not clear
Response: We have supplemented the sample size in the abstract: n=10.
Q2. Please add illustrated diagram for the method used to prepare PCL-Geltin-AgNPs composite fibers and provide the CAS number for the utilized materials.
Response: Thank you for your suggestion. The method diagram for preparing PCL Geltin AgNPs composite fiber was as follows, and the CAS number of the utilized materials was added in the article.
Q3. Add suitable references at the end of line 88
Response: Thank for your comments, we have added suitable references to the article.
Results:
Q4. In Fig. 6B please add a labelling for the 4 plates and in the figure legend add the description of the four plates.
Response: We have supplemented the description in Figure 6B as follows: It can be seen that under the same dilution factor, the number of colonies decreases to varying degrees with the increase of AgNPs content in fibers.
Q5. In Fig. 7A also please add a proper labelling for the 4 Confocal microscopy images and in the figure legend to clarify the difference between them.
Response: We have made the following changes to the legend of Figure 7a: Staining plot of live dead bacteria growing on the surface of PCL-Geltin-AgNPs in amounts of 0%, 1%, 3%, and 5% for 2 days, with the live bacteria in green and the dead bacteria in red.
Q6. Please expand the discussion section to include comparing the obtained results to up to date research work of using the bioactive glasses in orthodontics for protecting and remineralizing enamel.
Response: Thanks for your kind suggestion. According to the reviewers’ comment, we have added a description of bioactive glass to the discussion section as follows: If the use of antibacterial materials can be combined with materials that promote enamel remineralization, the incidence of enamel demineralization is greatly reduced. In recent years, a bioactive glass for dentistry has been developed and applied to numerous studies to remineralize WSL [1-3]. The use of bioactive glass has shown more effective and rapid enhancement of tooth enamel remineralization than milk-protein-derived casein phosphopeptide-amorphous calcium phosphate (CPP-ACP) [4].
Reference:
- Milly, H.; Festy, F.; Watson, T.F.; Thompson, I.; Banerjee, A. Enamel white spot lesions can remineralise using bio-active glass and polyacrylic acid-modified bio-active glass powders. Journal of dentistry 2014, 42, 158-166, doi:10.1016/j.jdent.2013.11.012.
- Bakry, A.S.; Marghalani, H.Y.; Amin, O.A.; Tagami, J. The effect of a bioglass paste on enamel exposed to erosive challenge. Journal of dentistry 2014, 42, 1458-1463, doi:10.1016/j.jdent.2014.05.014.
- Bakry, A.S.; Takahashi, H.; Otsuki, M.; Tagami, J. Evaluation of new treatment for incipient enamel demineralization using 45S5 bioglass. Dental materials : official publication of the Academy of Dental Materials 2014, 30, 314-320, doi:10.1016/j.dental.2013.12.002.
- Taha, A.A.; Patel, M.P.; Hill, R.G.; Fleming, P.S. The effect of bioactive glasses on enamel remineralization: A systematic review. Journal of dentistry 2017, 67, 9-17, doi:10.1016/j.jdent.2017.09.007.

Reviewer 3 Report
As described by the authors, demineralization around brackets is a relatively common complication in orthodontics. The development of adhesives with antibacterial properties, that reduce the incidence of this complication, would certainly have positive implications. Because of this, the current study is on a topic of relevance and general interest. On the one hand, I found the paper to be overall well written and much of it to be well described. However, there are some points that should be improved in the paper:
- Line 24 “ARIvalues”, the acronym should be explicit-
- The terms "leukoplakia" and "white spot lesion" refer to different clinical pictures. The sentence needs to be changed.
- In the introduction, make clear what is the purpose of the research and the possible null hypothesis.
- Line 178. It is not clear how polishing with "silicon ions" takes place, detailing the procedure, and possibly describing the materials used.
- Motivate the parameters chosen for the preparation of the SBS and ARI test. Are 40 specimens considered sufficient to have statistical significance? The same for the 10 chosen for the ARI test. In this regard, the estimate of the sample size is missing in the materials and methods. Is one month of soaking enough to reproduce clinical use?
- Line 194-200. The test carried out to evaluate the adhesion is described in the text as Shear Test (SBS), however in the section of materials and methods which refers to the description of the procedure it seems that reference is made to a Tensile Test (TBS). Clarify which test was performed.
- Table 1. It would be useful to add an indication of the statistical differences between the various groups (eg letters). In the results the authors state that there are no significant differences between the groups (line 318), in the discussion they state: "although the SBS of orthodontic adhesives decreased slightly with the increase of composite fibers (line 387). Clarify this aspect.
- Line 318. “Studies have shown that the shear-resistant bonding strength required for orthodontic brackets should be 5.9-7.8 MPa” The adhesion values strongly depend on how the test is set up (e.g. type of test, adhesion surface, aging) and are not directly comparable to each other. In my opinion, the inclusion of this information is superfluous, especially in the results section. To compare two adhesion values, it must at least be verified that the tests and the main parameters with which they are set up are the same. It is more appropriate to comment on the differences between the control and test group, since the test setup is the same. It is reiterated that the sample size must be sufficient to have statistical significance.
- Table 2. The same as Table 1.It would be useful to add an indication of the statistical differences between the various groups (eg letters).
Author Response
Dear reviewer,
Thanks a lot for your kind comments on our manuscript. These comments are valuable and very helpful for revising and improving our manuscript. In what follows, we would like to answer the questions you mentioned and give a detailed account of the changes made to the original manuscript.
Q1. Line 24 “ARI values”, the acronym should be explicit-
Response: Thank for your comments. We have explained the ARI abbreviation in lines 20-21, so we have not repeated them in line 24.
Q2. The terms "leukoplakia" and "white spot lesion" refer to different clinical pictures. The sentence needs to be changed.
Response: We are very sorry for our miswriting. We have changed the “leukoplakia” to “enamel demineralization” in the article.
Q3. In the introduction, make clear what is the purpose of the research and the possible null hypothesis.
Response: Thank you for pointing out this problem in our manuscript. We have further elaborated on the purpose of the study in the abstract section: In this study, a novel orthodontic adhesive containing polycaprolactone-gelatin-silver nanoparticles (PCL-Gelatin-AgNPs) composite fibers was prepared to prevent enamel demineralization in the process of orthodontic treatment.
In addition, we added the null hypothesis in the introduction: The first null hypothesis of this study is that there was no difference in antimicrobial between the experimental and control groups. In addition, the second null hypothesis is that there is no difference in the bonding performance of traditional orthodontic bonding and new adhesives. Finally, the third null hypothesis is that there is no significant difference in the ARI index between the different groups.
Q4. Line 178. It is not clear how polishing with "silicon ions" takes place, detailing the procedure, and possibly describing the materials used.
Response: We gratefully appreciate for your valuable suggestions. We have added the following methods to clean the tooth surface: Before bracket bonding, brush the tooth thoroughly with fluoride-free toothpaste, rinse under running water, and polish with a silicone polisher and low-speed hand piece for 10s.
Q5. Motivate the parameters chosen for the preparation of the SBS and ARI test. Are 40 specimens considered sufficient to have statistical significance? The same for the 10 chosen for the ARI test. In this regard, the estimate of the sample size is missing in the materials and methods. Is one month of soaking enough to reproduce clinical use?
Response: Thank you very much for the positive comments and constructive suggestions. We reviewed the literature, in which many scholars rated the sample size of each group as 10 when assessing bond strength and ARI index [11-13]{!!! INVALID CITATION !!! {Yu`, 2017 `#300}, #0;Yu, 2017 #300}, so this sample size was also selected in this study. We have supplemented the description of the sample size in the article as follows: From October 2020 to February 2021, a total of 40 premolars with intact enamel and standard color were selected, ten teeth were tested for each group.
Fixed appliances stay in the mouth for a long time, and this experiment only tested the bond strength for 1 month, and it is difficult to evaluate the long-term adhesion performance of the adhesive. However, the results of this 1-month period were basically not different from those of the control group, so we considered it to have the same performance as normal bonding agents. Since in vitro evaluation of bond strength does not consider environmental factors such as abrasion and chewing force [14], average adhesion measurements may be of limited value in explaining in vitro bond strength. To simulate the intraoral environment, we will subsequently conduct animal experiments to evaluate the performance of the adhesive in the mouth.
Q6. Line 194-200. The test carried out to evaluate the adhesion is described in the text as Shear Test (SBS), however in the section of materials and methods which refers to the description of the procedure it seems that reference is made to a Tensile Test (TBS). Clarify which test was performed.
Response: Thanks very much for your comment, which is highly appreciated. We further examined the experimental process of the article, clarified that the research done in this experiment is tensile bond strength (TBS) testing, and modified the relevant content in the article.
Q7. Table 1. It would be useful to add an indication of the statistical differences between the various groups (eg letters). In the results the authors state that there are no significant differences between the groups (line 318), in the discussion they state: "although the SBS of orthodontic adhesives decreased slightly with the increase of composite fibers (line 387). Clarify this aspect.
Response: Thanks for your kind question, we have added P values for each experimental group compared to the control group in Table 1.
We've made the following changes to the discussion section: In this study, although the orthodontic adhesive TBS decreased slightly with the increase of composite fibers, no significant difference was found when the experimental group was compared with the control group. The TBS values in each group were higher than 5.9 MPa and lower than 13.5 Mpa, all of which met the clinical adhesion criteria.
Q8. Line 318. “Studies have shown that the shear-resistant bonding strength required for orthodontic brackets should be 5.9-7.8 MPa” The adhesion values strongly depend on how the test is set up (e.g., type of test, adhesion surface, aging) and are not directly comparable to each other. In my opinion, the inclusion of this information is superfluous, especially in the results section. To compare two adhesion values, it must at least be verified that the tests and the main parameters with which they are set up are the same. It is more appropriate to comment on the differences between the control and test group, since the test setup is the same. It is reiterated that the sample size must be sufficient to have statistical significance.
Response: Thanks for your kind suggestion. According to the reviewers’ comment, we have removed the “Studies have shown that the shear-resistant bonding strength required for orthodontic brackets should be 5.9-7.8 MPa” in the results section.
Q9. Table 2. The same as Table 1. It would be useful to add an indication of the statistical differences between the various groups (eg letters).
Response: Thanks very much for your comment, which is highly appreciated. We have supplemented the table with a description of statistical differences.
Reference:
- Yu, F.; Dong, Y.; Yu, H.H.; Lin, P.T.; Zhang, L.; Sun, X.; Liu, Y.; Xia, Y.N.; Huang, L.; Chen, J.H. Antibacterial Activity and Bonding Ability of an Orthodontic Adhesive Containing the Antibacterial Monomer 2-Methacryloxylethyl Hexadecyl Methyl Ammonium Bromide. Scientific reports 2017, 7, 41787, doi:10.1038/srep41787.
- Liu, Y.; Zhang, L.; Niu, L.N.; Yu, T.; Xu, H.H.K.; Weir, M.D.; Oates, T.W.; Tay, F.R.; Chen, J.H. Antibacterial and remineralizing orthodontic adhesive containing quaternary ammonium resin monomer and amorphous calcium phosphate nanoparticles. Journal of dentistry 2018, 72, 53-63, doi:10.1016/j.jdent.2018.03.004.
- Zhang, N.; Weir, M.D.; Romberg, E.; Bai, Y.; Xu, H.H. Development of novel dental adhesive with double benefits of protein-repellent and antibacterial capabilities. Dental materials : official publication of the Academy of Dental Materials 2015, 31, 845-854, doi:10.1016/j.dental.2015.04.013.
- Pickett, K.L.; Sadowsky, P.L.; Jacobson, A.; Lacefield, W. Orthodontic in vivo bond strength: comparison with in vitro results. The Angle orthodontist 2001, 71, 141-148, doi:10.1043/0003-3219(2001)071<0141:Oivbsc>2.0.Co;2.

Reviewer 4 Report
Manuscript of considerable interest for the orthodontic sector, a major revision is required before proceeding with the evaluation of any publication.
Abstract, highlight more the data reported with the statistically significant results.
Keywords; add specifics, these are too generic.
Introduction: add the studies carried out by the study group of Prof Lanteri Valentina et al, on pre-bonding prophylaxis systems with relative detachment tests, and the treatments of lesions (such as post-orthodontic white spots) using biomimetic hydroxyapatite by Prof Scribante et al.
10.3390/ijerph19148676 10.4103/2278-0203.197392 International Journal of Clinical DentistryVolume 7, Issue 2, Pages 191 - 1971 January 2014Pre-bonding prophylaxis and brackets detachment: An experimental comparison of different methods
Materials and methods well described, how was the sample size calculated?
Results, very confusing, to reorganize the data, and high resolution graphs clearly visible, beautiful images at the same time.
Discussion add as future goals the use of hydroxyapatite and evaluate the antibacterial action of the zinc it contains.
Conclusions, draw up a linear workflow through proactive action.
Bibliography; add references required
Author Response
Dear reviewer,
Thanks a lot for your kind comments on our manuscript. These comments are valuable and very helpful for revising and improving our manuscript. In what follows, we would like to answer the questions you mentioned and give a detailed account of the changes made to the original manuscript.
Q1. Abstract, highlight more the data reported with the statistically significant results.
Response: Thanks for your constructive suggestion, which is highly appreciated. In the abstract, we added descriptions of statistically significant experimental data. The addition of complex antimicrobial fibers to 3M Transbond XT adhesive can significantly reduce the CFU of bacterial biofilms (P < 0.05). The bacterial survival rate on the surface of the specimen decreased with the increase of PCL-Geltin-AgNPs short fibers (P < 0.05).
Q2. Keywords: add specifics, these are too generic.
Response: Thank you for your comment. We've made the following adjustments to the keywords of the article: PCL-Gelatin-AgNPs fibers; orthodontic adhesive; antibacterial property; tensile bonding strength; Streptococcus mutans
Q3. Introduction: add the studies carried out by the study group of Prof Lanteri Valentina et al, on pre-bonding prophylaxis systems with relative detachment tests, and the treatments of lesions (such as post-orthodontic white spots) using biomimetic hydroxyapatite by Prof Scribante et al.
10.3390/ijerph19148676 10.4103/2278-0203.197392
International Journal of Clinical Dentistry
Volume 7, Issue 2, Pages 191 - 1971 January 2014
Pre-bonding prophylaxis and brackets detachment: An experimental comparison of different methods
Response: Thank you very much for your recommendation. We have added the following to the introduction: The use of fluorine-based products has been the gold standard for hard tissue remineralization due to its ability to integrate into tooth enamel to convert hydroxyapatite from fluoroapatite. However, Lanteri et al. found that the use of pre-orthodontic treatment of fluoride paint or varnish produces lower bond strength, which may increase the risk of brackets falling off [1].
In the discussion section, we made the following additions:In recent years, Scribante et al. have used biomimetic nano-hydroxyapatite to induce visible tooth enamel remineralization after 2 application cycles, and the bond strength can be considered clinically acceptable [2].
Q4. Materials and methods well described, how was the sample size calculated?
Response: Thanks for your kind question. We reviewed the literature, in which many scholars rated the sample size of each group as 10 when assessing bond strength and ARI index [3-5]{!!! INVALID CITATION !!! {Yu`, 2017 `#300}, #0;Yu, 2017 #300}, so this sample size was also selected in this study.
Q5. Results, very confusing, to reorganize the data, and high resolution graphs clearly visible, beautiful images at the same time.
Response: Thanks for your constructive suggestion, which is highly appreciated. We have carefully scrutinized the manuscript, and made corresponding revisions including some grammatical errors and long sentences, etc. In addition, we modified the description of some experimental results.
Q6. Discussion add as future goals the use of hydroxyapatite and evaluate the antibacterial action of the zinc it contains.
Response: Thanks for your suggestion, related research on zinc-hydroxyapatite is now included in the revised manuscript as follows: In addition, some scholars have added zinc-hydroxyapatite with remineralization to toothpaste and resin composite, both of which show good enamel protection [6-8]. Therefore, the combined application of antimicrobial substances and remineralized materials is a strategy to prevent WSL.
Q7. Conclusions, draw up a linear workflow through proactive action.
Response: Thanks a lot for you suggestion, we have drawn a illustrated diagram.
Q8. Bibliography: add references required
Response: We have further supplemented the references in the article.
Reference:
- Lanteri, V.; Segù, M.; Doldi, J.; Butera, A. Pre-bonding prophylaxis and brackets detachment: An experimental comparison of different methods. International Journal of Clinical Dentistry 2014, 7, 191-197.
- Scribante, A.; Dermenaki Farahani, M.R.; Marino, G.; Matera, C.; Rodriguez, Y.B.R.; Lanteri, V.; Butera, A. Biomimetic Effect of Nano-Hydroxyapatite in Demineralized Enamel before Orthodontic Bonding of Brackets and Attachments: Visual, Adhesion Strength, and Hardness in In Vitro Tests. BioMed research international 2020, 2020, 6747498, doi:10.1155/2020/6747498.
- Yu, F.; Dong, Y.; Yu, H.H.; Lin, P.T.; Zhang, L.; Sun, X.; Liu, Y.; Xia, Y.N.; Huang, L.; Chen, J.H. Antibacterial Activity and Bonding Ability of an Orthodontic Adhesive Containing the Antibacterial Monomer 2-Methacryloxylethyl Hexadecyl Methyl Ammonium Bromide. Scientific reports 2017, 7, 41787, doi:10.1038/srep41787.
- Liu, Y.; Zhang, L.; Niu, L.N.; Yu, T.; Xu, H.H.K.; Weir, M.D.; Oates, T.W.; Tay, F.R.; Chen, J.H. Antibacterial and remineralizing orthodontic adhesive containing quaternary ammonium resin monomer and amorphous calcium phosphate nanoparticles. Journal of dentistry 2018, 72, 53-63, doi:10.1016/j.jdent.2018.03.004.
- Zhang, N.; Weir, M.D.; Romberg, E.; Bai, Y.; Xu, H.H. Development of novel dental adhesive with double benefits of protein-repellent and antibacterial capabilities. Dental materials : official publication of the Academy of Dental Materials 2015, 31, 845-854, doi:10.1016/j.dental.2015.04.013.
- Colombo, M.; Beltrami, R.; Rattalino, D.; Mirando, M.; Chiesa, M.; Poggio, C. Protective effects of a zinc-hydroxyapatite toothpaste on enamel erosion: SEM study. Annali di stomatologia 2016, 7, 38-45, doi:10.11138/ads/2016.7.3.038.
- Poggio, C.; Gulino, C.; Mirando, M.; Colombo, M.; Pietrocola, G. Protective effect of zinc-hydroxyapatite toothpastes on enamel erosion: An in vitro study. Journal of clinical and experimental dentistry 2017, 9, e118-e122, doi:10.4317/jced.53068.
- Li, Y.; Zhang, D.; Wan, Z.; Yang, X.; Cai, Q. Dental resin composites with improved antibacterial and mineralization properties via incorporating zinc/strontium-doped hydroxyapatite as functional fillers. Biomedical materials (Bristol, England) 2022, 17, doi:10.1088/1748-605X/ac6b72.

Reviewer 5 Report
Thank you for the opportunity to review this paper. This is an exciting work, considering that it aims to provide a new idea and method for preventing enamel demineralization around the brackets of fixed appliances, which has clinical practical value and application prospects. Despite the hard work collecting data in this manuscript, I do not think this present version is good enough to be published in the JFB
· The introduction section is very vague and shallow. The authors claim …“we synthesized a material with excellent biocompatibility”… but it seems that the concept of biocompatibility is very precarious; they only performed primary tests; there are no animal model tests or clinical trials. What is the correlation between well plates and complex clinical reality (so be humble)
· In the M&M section, describe the n for all kinds of testing.
o Did you measure the roughness before to assess the contact angle? If not why?
o What was the degree of conversion of the adhesive before and after the fibre was mixed?
· In the Discussion section, some part should be in the introduction (l. 359-360).
o What was the mechanism of Ag-ion releasing from the adhesive?
o How would the Ag-ion release affect the patient, especially the young ones?
o Discuss the flaws and how they were overcome is welcome to help future researchers in this field.
Author Response
Dear reviewer,
Thanks a lot for your kind comments on our manuscript. These comments are valuable and very helpful for revising and improving our manuscript. In what follows, we would like to answer the questions you mentioned and give a detailed account of the changes made to the original manuscript.
Q1. The introduction section is very vague and shallow. The authors claim “we synthesized a material with excellent biocompatibility” but it seems that the concept of biocompatibility is very precarious; they only performed primary tests; there are no animal model tests or clinical trials. What is the correlation between well plates and complex clinical reality (so be humble)
Response: Thank the reviewer for the suggestion. We have checked and revised the wording in the article and further refined the introduction section.
Q2. In the M&M section, describe the n for all kinds of testing.
Response: Thanks for your kind suggestion. According to the reviewers’ comment, We supplemented the sample size in the M&M section: Each group produced 10 specimens for subsequent bond strength and ARI index evaluation.
Q3. Did you measure the roughness before to assess the contact angle? If not why?
Response: Thanks for your kind question. In this experiment, roughness was not evaluated before detecting the contact angle. Through literature review, we understand that the contact angle of solid surfaces is affected by physical and chemical factors, including chemical composition, surface texture (roughness), and surface chemistry (heterogeneity) [1,2]. The fiber surface roughness is related to the receiver used in the electrospinning process [1]. In this experiment, aluminum foil was used as the receiving device, so it had little effect on roughness. Therefore, the fiber roughness is not tested.
Q4. What was the degree of conversion of the adhesive before and after the fiber was mixed?
Response: Thanks for your kind question. Through preliminary literature review, we learned that the conversion rate of resin is related to the chemical and mechanical properties of the polymer, such as diameter tensile strength, compressive strength with hardness, flexural modulus and strength [3,4]. Therefore, we conducted pre-experiments on the conversion rate of the bonding agent prior to performing bond strength tests. The results are shown in Figure C. There was no statistical difference in DC between the control and short fibre incorporation groups. (P>0.05) However, in the pre-experiment we used a small sample size (n=3) and we were unsure if increasing the sample size would have an effect on the experimental results, so we did not mention it in the article.
Figure 4. Degree of monomer conversion of all materials after light curing for 10 s.
Q5. In the Discussion section, some part should be in the introduction (l. 359-360).
Response: Thanks for your constructive suggestion. We have made changes to the discussion section.
Q6. What was the mechanism of Ag-ion releasing from the adhesive?
Response: Thanks for your kind question. The small particle size and large surface area of AgNPs allows them to release more Ag+ at low filler levels, thereby reducing the concentration of Ag+ required for efficacy [5,6]. Li et al. found that adhesives containing AgNPs inhibited not only Streptococcus mutans on their surface, but also Streptococcus mutans suspended in the medium away from their surface [7]. This suggests that resins containing AgNPs are long-range lethal, possibly due to the release of Ag+ [8]. Previous studies have shown that Ag-containing resin composites have long-lasting antimicrobial activity due to the continuous release of Ag+ [9]. Yoshida et al. found that Ag-containing dental composites were shown to inhibit the growth of Streptococcus mutans when tested for 6 months [10].
Morones et al. showed that the bactericidal effect of nanosilver is related to the release of Ag+. The data suggest that rapid Ag+ release occurs when the nanoparticles are first dissolved, but only at level of <5 µm. No further dissolution occurs and the free Ag+ concentration decreases, possibly due to reduction processes to form Ag+ containing clusters or re-association with the original nanoparticles. This analysis corroborated the presence of micro-molar concentrations of silver ions, which will have a contribution to the biocidal action of the silver nanoparticles [8].
Q7. How would the Ag-ion release affect the patient, especially the young ones?
Response: Silver nanoparticles have been demonstrated to be excellent antimicrobial agents in the biomedical field in a wide variety of microorganisms [11], even for oral bacteria [12]. The action mechanism of AgNPs against bacteria has been widely described. Factors such as ion release capacity, biofunctionalization, zeta potential, size, shape, and other physicochemical properties could permit the adhesion of silver ions/particles around of cell wall of bacterial cells causing cell wall disruption and facilitating the silver penetration into the cell attaching to thiols, amino, and hydroxyl groups affecting DNA replication, proteins production, promotion of apoptosis, and, finally, leading cell death.
Most of the patients involved in orthodontic treatment are young people. Orthodontic adhesives containing silver ions can control the aggregation of Streptococcus mutans on the surface of orthodontic appliances, thereby limiting the occurrence of enamel demineralization during orthodontics, which has a positive effect on aesthetics.
The Ag-containing bonding agent inhibited not only the S. mutans on its surface, but also the S. mutans away from its surface suspended in the culture medium. This indicates that Nag containing resin has a long-distance killing capability, likely due to the release of Ag ions [1].
Reference:
- Alam, A.; Ewaldz, E.; Xiang, C.; Qu, W.; Bai, X. Tunable Wettability of Biodegradable Multilayer Sandwich-Structured Electrospun Nanofibrous Membranes. Polymers 2020, 12, doi:10.3390/polym12092092.
- Szewczyk, P.K.; Ura, D.P.; Metwally, S.; Knapczyk-Korczak, J.; Gajek, M.; Marzec, M.M.; Bernasik, A.; Stachewicz, U. Roughness and Fiber Fraction Dominated Wetting of Electrospun Fiber-Based Porous Meshes. Polymers 2018, 11, doi:10.3390/polym11010034.
- Yang, S.Y.; Han, A.R.; Kim, K.M.; Kwon, J.S. Acid neutralizing and remineralizing orthodontic adhesive containing hydrated calcium silicate. Journal of dentistry 2022, 123, 104204, doi:10.1016/j.jdent.2022.104204.
- Eliades, T.; Eliades, G.; Brantley, W.A.; Johnston, W.M. Polymerization efficiency of chemically cured and visible light-cured orthodontic adhesives: degree of cure. American journal of orthodontics and dentofacial orthopedics : official publication of the American Association of Orthodontists, its constituent societies, and the American Board of Orthodontics 1995, 108, 294-301, doi:10.1016/s0889-5406(95)70024-2.
- Cheng, L.; Weir, M.D.; Xu, H.H.; Antonucci, J.M.; Kraigsley, A.M.; Lin, N.J.; Lin-Gibson, S.; Zhou, X. Antibacterial amorphous calcium phosphate nanocomposites with a quaternary ammonium dimethacrylate and silver nanoparticles. Dental materials : official publication of the Academy of Dental Materials 2012, 28, 561-572, doi:10.1016/j.dental.2012.01.005.
- Cheng, Y.J.; Zeiger, D.N.; Howarter, J.A.; Zhang, X.; Lin, N.J.; Antonucci, J.M.; Lin-Gibson, S. In situ formation of silver nanoparticles in photocrosslinking polymers. Journal of biomedical materials research. Part B, Applied biomaterials 2011, 97, 124-131, doi:10.1002/jbm.b.31793.
- Li, F.; Weir, M.D.; Chen, J.; Xu, H.H. Comparison of quaternary ammonium-containing with nano-silver-containing adhesive in antibacterial properties and cytotoxicity. Dental materials : official publication of the Academy of Dental Materials 2013, 29, 450-461, doi:10.1016/j.dental.2013.01.012.
- Morones, J.R.; Elechiguerra, J.L.; Camacho, A.; Holt, K.; Kouri, J.B.; Ramírez, J.T.; Yacaman, M.J. The bactericidal effect of silver nanoparticles. Nanotechnology 2005, 16, 2346-2353, doi:10.1088/0957-4484/16/10/059.
- Damm, C.; Münstedt, H.; Rösch, A. Long-term antimicrobial polyamide 6/silver-nanocomposites. Journal of Materials Science 2007, 42, 6067-6073, doi:10.1007/s10853-006-1158-5.
- Yoshida, K.; Tanagawa, M.; Atsuta, M. Characterization and inhibitory effect of antibacterial dental resin composites incorporating silver-supported materials. Journal of biomedical materials research 1999, 47, 516-522, doi:10.1002/(sici)1097-4636(19991215)47:4<516::aid-jbm7>3.0.co;2-e.
- Jasso-Ruiz, I.; Velazquez-Enriquez, U.; Scougall-Vilchis, R.J.; Morales-Luckie, R.A.; Sawada, T.; Yamaguchi, R. Silver nanoparticles in orthodontics, a new alternative in bacterial inhibition: in vitro study. Progress in orthodontics 2020, 21, 24, doi:10.1186/s40510-020-00324-6.
- Pérez-Díaz, M.A.; Boegli, L.; James, G.; Velasquillo, C.; Sánchez-Sánchez, R.; Martínez-Martínez, R.E.; Martínez-Castañón, G.A.; Martinez-Gutierrez, F. Silver nanoparticles with antimicrobial activities against Streptococcus mutans and their cytotoxic effect. Materials science & engineering. C, Materials for biological applications 2015, 55, 360-366, doi:10.1016/j.msec.2015.05.036.

Round 2
Reviewer 1 Report
This reviewer satisfied your sincere responses and enough revising the manuscript.
Reviewer 4 Report
the manuscript has been properly revised, it can be published
Reviewer 5 Report
Congratulations for the good work.